# Cancer of the Cervix in Bulgaria: Epidemiology of a Crisis

**DOI:** 10.3390/healthcare11030318

**Published:** 2023-01-20

**Authors:** Angel Yordanov, Mariela Vasileva-Slaveva, Noya Galai, David Faraggi, Milan Paul Kubelac, Irina Tripac-Iacovleva, Neville Calleja, Riccardo Di Fiore, Jean Calleja-Agius

**Affiliations:** 1Department of Gynecological Oncology, Medical Universiy Pleven, 5800 Pleven, Bulgaria; 2Department of Breast Surgery, “Dr. Shterev” Hospital, 1330 Sofia, Bulgaria; 3Research Institute, Medical University Pleven, 5800 Pleven, Bulgaria; 4Bulgarian Breast and Gynecological Cancer Association, 1784 Sofia, Bulgaria; 5Department of Statistics, University of Haifa, Haifa 31905, Israel; 6Department of Medical Oncology, The Oncology Institute “Prof. Dr. Ion Chiricuţă”, 400015 Cluj-Napoca, Romania; 7Department of Oncology, “Iuliu Hatieganu” University of Medicine and Pharmacy, 400347 Cluj-Napoca, Romania; 8Department of Gynaecology, The Oncology Institute of Moldova, MD 2025 Chisinau, Moldova; 9Directorate for Health Information and Research, PTA 1313 Tal-Pietà, Malta; 10Department of Anatomy, Faculty of Medicine and Surgery, University of Malta, MSD 2080 Msida, Malta; 11Sbarro Institute for Cancer Research and Molecular Medicine, Center for Biotechnology, College of Science and Technology, Temple University, Philadelphia, PA 19122, USA

**Keywords:** cancer of the uterine cervix, survival, screening programs, human papillomavirus (HPV)

## Abstract

Eastern Europe continues to have the highest rates of cancer of the uterine cervix (CUC) and human papillomavirus (HPV) infection in Europe. Aim: The aim of this study was to investigate CUC trends in Bulgaria in the context of a lack of a population-based screening program and a demographic crisis. Methodology: This was a retrospective study of 7861 CUC patients who were registered in the Bulgarian National Cancer Registry (BNCR) between 2013 and 2020 and followed up with until March 2022. We used descriptive statistics and modeling to assess temporal trends in new CUC incidence rates and identify factors associated with survival. Results: Bulgaria’s population has decreased by 11.5% between 2011 and 2021. The CUC incidence rate decreased from 29.5/100,000 in 2013 to 23.2/100,000 in 2020 but remains very high. The proportion of patients diagnosed in earlier stages of CUC has decreased over time. Up to 19% of patients with CUC in Bulgaria are diagnosed between the age of 35 and 44 years. The median survival was 101.5 months, with some improvement in later years (adjusted HR = 0.83 for 2017–2020). Conclusions: In countries with well-established population-based screening, CUC is nowadays considered a rare disease. However, it is not considered rare in Bulgaria. Population-based screening starting at an earlier age is the fastest way to improve outcomes.

## 1. Introduction

Cancer of the uterine cervix (CUC) ranks fourth in both incidence and mortality rate among all oncological diseases in women globally [1]. In 2020, worldwide, there were over 604,000 women diagnosed with CUC and over 340,000 deaths registered due to this cancer [2]. This represents a 5.7% increase in the new incidence rate and an 8.9% increase in mortality since 2018 [1]. It is expected that by 2040, the number of new cases will start to decrease in Europe [3]. However, Eastern Europe remains the region with the highest CUC morbidity to date [1].

It is well known that there are huge disparities in cancer incidence, management and survival between and within countries [4]. The survival gap in Europe is biggest between patients from Eastern Europe and other parts of Europe [5]. In Bulgaria, according to the GLOBOCAN database, the estimated age-standardized incidence rate of CUC for 2020 was 18.3 per 100,000 [6], which was among the highest in Europe. Bulgaria also has the third-highest age-standardized mortality [1] with a five-year survival rate that is at least 10% lower than the survival rate of patients in Western countries [5].

In total, 98.7% of all cases of CUC can be related to human papillomavirus (HPV) infection [7,8]. This association raises the opportunity for effective prevention of the disease through HPV vaccination. In 2018, the World Health Organization (WHO) announced the “Call for Action” for the eradication of CUC as a public health problem [9]. It has been estimated that in the next 50 years, over 44 million women will be diagnosed with CUC if no actions are applied in primary and secondary prevention [10]. WHO recommendations include vaccinating girls against HPV, starting between the ages of 9 and 14; conducting screening examinations with an HPV test for women above the age of 30 (or above the age of 25 in cases with proven infection); and providing treatment of invasive cancer in a timely manner at any age [11]. The burden of HPV infection in different countries falls into a wide range and differs among women with negative cytology, precancerous lesions and invasive cancer [12]. According to the report of the HPV center of the International Agency for Research on Cancer (IARC) in 2021, while there is no available data for Bulgaria, the HPV burden in Eastern Europe indicated that 9.7% of women had been carriers of HPV 16/18 infection at some point in their life [12]. These are exactly the same subtypes of HPV that are related to 84.7% of invasive CUC. An HPV vaccination program was introduced in Bulgaria in 2017 [12]. In 2019, 6.0% of the girls in the targeted population had received the first dose. Thereafter, this percentage of girls receiving the first dose decreased annually [12]. However, there is still no population-based national screening program for CUC in Bulgaria [12].

The aim of the study was to investigate the most recent real-world data on the epidemiology, prognosis and survival of CUC in Bulgaria against the background of the country possessing no population-based screening program.

## 2. Materials and Methods

This is a retrospective population-wide study of all patients with CUC registered in the Bulgarian National Cancer Registry (BNCR) between 2013 and 2020. Patients were followed up with until March 2022 with information on the stage of the cancer diagnosis, treatment received and mortality. Information was obtained from BNCR after official permission. 

### 2.1. Study Population

For the study period of 2013 to 2020, a total of 7861 women were registered in BNCR with CUC. Patient groups, according to the number and sequence of cancer diagnosis, are shown in Figure 1.

### 2.2. Data and Statistical Analysis

Incidence rates of CUC were calculated by the year of diagnosis and by the age group as the number of new cases divided per 100,000. The temporal trend in the incidence rates was assessed with a linear model, with weights reflecting the size of the population in each age group. Overall survival was defined as the time, in months, between the date of the patient’s first diagnosis of CUC and the date of death from any cause by 31 March 2022. We used a Kaplan–Meier analysis to estimate the overall survival in patient groups defined by age and the time period (defined as early for 2013–2016, or late for 2017–2020), as well as the stage, grade and histological type of the cancer at diagnosis. A log rank test was used to test for the significance of differences in survival between groups. A *p*-value ≤ 0.05 was considered statistically significant.

For the survival analysis, we excluded Group 3, six patients from Group 2 who had multiple cancer diagnosis on the same day of the cervical cancer diagnosis, and one patient who did not follow up. These exclusions resulted in a reduced sample size of 7486. However, there were 218 patients who were diagnosed on the day of death; these did not contribute to the survival time estimate. Therefore, the final number of patients for the survival analysis after all the exclusions was 7268. All analyses were performed with STATA (StataCorp. 2021. Stata Statistical Software: Release 17).

## 3. Results

### 3.1. Decline in the General Population in Bulgaria

The population in Bulgaria has been constantly decreasing over the last few years [13]. At the same time, life expectancy was found to increase up to 2020 [14]. In 2021, there was a decrease in life expectancy in all EU member states due to the Coronavirus disease 2019, with the biggest decrease observed in Bulgaria [15]. According to the database we analyzed from BNCR, the population of women in Bulgaria decreased by almost 200,000 between 2013 and 2020. Additionally, the age distribution of the population shifted towards a higher proportion of the older age group while the size of younger age groups decreased. Figure 2 shows the decrease in the population and the change in the distribution by age over time (Figure 2A,B).

### 3.2. CUC Burden in Bulgaria 2013–2020

There was an overall decrease in the rates of new cases of CUC in Bulgaria from 29.5/100,000 in 2013 to 23.2/100,000 in 2020 (Figure 3). The average decline was −0.97/100,000 per year; *p* < 0.001. Despite that, on average, 982 patients with CUC are diagnosed in Bulgaria annually.

The stage at diagnosis was known for 7050 (89.7%) of all patients registered in the BNCR for the study period. The age and stage distribution of new cases is shown in Figure 4. The stated percentage is the percentage of women in this age category from all patients. A total of 26.3% of all women were over 65 years of age at the time of diagnosis (Figure 4).

The percentage of patients diagnosed in earlier stages decreased in the last years of the study. For example, the proportion diagnosed at Stage I decreased from 48.3% in 2015 to 37.5% in 2020. In addition, the older the patient is at diagnosis, the higher her risk is of being diagnosed with an advanced disease (Figure 5). Among the very young patients aged between 15 and 29 years, almost 63% were diagnosed at Stage I. The percentage of patients at Stage I who were over 65 years at diagnosis was two times lower (33%) compared to the younger patients (Figure 4).

### 3.3. Survival

The median overall survival of all 7642 patients in the study (excluding those diagnosed only on a death certificate) was 92.5 months. For patients in Groups 1 and 2, the combined median survival was 101.3 months. This indicates that, as expected, those with prior cancer diagnoses had lower survival rates (92.5 vs. 101.3 months). The difference in survival according to the year of diagnosis (2013–2016 vs. 2017–2020) is shown in Figure 6. The difference is significant (*p* = 0.046), and it is in favor of more recently diagnosed women.

As prognostic factors for overall survival, we investigated the following: age at diagnosis; TNM stage at diagnosis; and tumor grade and histology within all patients in Groups 1 and 2 (Figure 7). All those factors were significantly associated with survival.

In a multivariate Cox regression model, factors that were significantly associated with survival with *p* < 0.001 included: age at diagnosis (Hazard ratio, HR = 1.03/year), stage (with Stage I as reference: Stage 2, HR = 1.93; Stage 3, HR = 3.16; Stage 4, HR = 8.01; stage unknown, HR = 3.07), and period of diagnosis: late (2017–2020) versus early (2013–2016) (HR = 0.83). Differences between Groups 1 and 2 were no longer significant after adjusting for all other factors.

## 4. Discussion

In Bulgaria, the CUC incidence rate decreased from 29.5/100,000 in 2013 to 23.2/100,000 in 2020 but remains very high. The proportion of patients diagnosed in earlier stages has decreased over time. The median overall survival was 101.5 months with some improvement in later years (adjusted HR = 0.83 for 2017–2020). Up to 19% of patients with CUC in Bulgaria are diagnosed between the age of 35 and 44 years, which shows the need for an earlier onset of the screening program, especially if it is HPV-based CUC.

### 4.1. Demographic Crisis

The population in Bulgaria is constantly decreasing. Between the last two national censuses in 2011 and 2021, the population decreased by 11.49%, or 844,781 people [16], which makes Bulgaria the fastest-shrinking nation in the world. This is mainly due to mass outbound migration, but this is not the only factor. The life expectancy at birth decreased in all countries in 2020, but within Europe, it decreased the most in Bulgaria [15]. This decrease in the population has a major impact on the new incidence rates of cancer.

### 4.2. Historical Data on Early Diagnosis of CUC in Bulgaria

Despite the lack of a national screening program for CUC in Bulgaria, there have been a few attempts to improve the early diagnosis. In 2009–2014, the program called “Stop and Check Yourself” [17] was financed by the European social fund and included activities for early diagnosis of colorectal, breast and cervical cancers. In total, 619,120 people have been invited for a screening examination, but only 55,898 exams were performed, of which 33,237 were for cervical cancer. Finally, 171 cervical cancers were diagnosed. As part of the program, a National Screening Registry was created, but it was functional only during the program period.

In 2014–2020 [18], and 2021–2025 [19], national programs for the prevention of non-communicable diseases were established in Bulgaria. The budget for screening examinations has been distributed through the National Insurance Fund. The Fund is paying EUR 5 for screening cytology and has an annual budget of EUR 2 million. In these programs, screening examinations should be initiated by the patients, and the HPV genotyping is covered out of pocket by the patient. During the first program period, additional cervical cancer awareness programs covered the PAP tests of 1297 women, and nine cases of dysplasia were found.

The results of all these attempts are shown in Figure 8. The highest coverage of CUC screening for the Bulgarian population according to Eurostat was about 35% and was observed in 2016.

Currently, Bulgaria is in the process of establishing an anticancer plan, but the process is not yet finalized [21]. Introducing a screening program for CUC is one of the aims of this plan. HPV testing is included as part of the screening methods. The CUC screening program is expected to start in 2025, if prioritized by the government at that point. According to the anticancer plan, HPV-based screening should not start before the age of 30. Between the age of 30 and 34, the authors of the plan see no evidence of the benefit of screening [21].

In France, the High Authority for Health (HAS) recently updated their evidence-based recommendations. They now include three yearly cytology screenings for women aged 25–29 years and HPV screening for those aged 30–65 years, with an extension of the screening interval to 5 years if the HPV test is negative [22]. A systematic review of screening guidelines in 11 countries across North America, Europe and Asia-Pacific compares the age of onset of screening. In the U.K., Australia and China, HPV genotyping starts at the age of 25 years. In some parts of Germany and in Italy, it starts at the age of 30, but testing with cytology is recommended from 20 years onwards in Germany and from 25 years in Italy. CUC screening in 8 of 11 countries ends at the age of 65; in Australia, it ends at the age group of 70–74; and in Germany and in Japan, there is no specific end date [23].

### 4.3. Burden of CUC

In 2020, according to GLOBOCAN, the new cases of CUC in Bulgaria were estimated to be 1009, and the new incidence rate was 18 per 100,000 [2]. Historically, Bulgaria has had one of the highest CUC rates in Europe [24]. According to the data from the BNCR, the registered cases in 2020 were only 822, which is lower than the GLOBOCAN estimation. The difference can be due to several reasons, but the most likely is that there is also a higher-than-expected decrease in the population. The estimated projection for the population in Bulgaria is 300,000 people more than what was observed in the last censuses in 2021 [25]. Another reason could be the problems in registering new cases in the BNCR [26].

### 4.4. HPV Vaccination Coverage

HPV vaccines are currently in the recommended vaccination list in Bulgaria. Unfortunately, the vaccination rate is below 6% for the first dose [12]. The study, using information from the Global Cancer Observatory and WHO/UNICEF database published in August 2021, estimated that the global HPV immunization coverage for 2018 was at 12.2%. In some high-income countries, the HPV complete vaccination has covered up to 69% of the population. [27] As of December 2021, most of the countries in the European Union and the European Economic Area already have, or are considering, moving from a girls-only HPV vaccination strategy to a universal, or gender-neutral, HPV vaccination strategy [28]. In the upper-middle-income countries (which include Bulgaria, according to the World Bank) the population coverage is 44% [27]. According to the European Cancer Organization, no EU country has achieved an uptake of 90% or more for the final dose [29]. Therefore, this strategy for reducing CUC burden, despite being very effective, will take a long time.

### 4.5. Prognostic Factors—Age and Stage

Only 5% of new cases of CUC in Bulgaria were found in women under the age of 35. Progression from HPV infection to invasive cancer can take between 2 and 10 years [30]. Therefore, HPV-based screening should start exactly in this age group and should include women of age 25–34 or at least 30–34. The group of patients with ages above 65 years made up 26.3% of all cases diagnosed in the study period; these patients tended to contract more advanced diseases (Figure 4). Population-based screening would also have a major effect in this group by allowing earlier detection and treatment. Between the youngest and oldest patients, the chance of being diagnosed with early cancer decreases almost by half (62% vs. 33%).

### 4.6. Prognostic Factors—Histology and Grade

HPV infection is mostly related to squamous cell carcinoma. The percentage of squamous cell carcinoma among our patients is higher than reported in the literature. This is probably due to the fact that the HPV infection rates in Eastern Europe are higher than in other parts of Europe [12]. With the introduction of a screening program, we expect that new incidence rates of this histological type will be reduced. 

### 4.7. Survival

According to the CONCORD 3 study, the 5-year relative overall survival in Bulgaria for patients diagnosed up to 2014 is 54.8% [5]. This is 10.6% less than in Belgium, 14.7% less than in Denmark and 18.5% less than in Norway. Based on the BNCR, the estimated 5-year survival was 56.2% (95% CI 54.9–57.4%) overall, which suggests very little improvement. We may need more data to adequately estimate the 5-year survival for those patients diagnosed in more recent years. In our multivariate model, after adjusting for age and stage, patients diagnosed after 2017 had better survival with HR = 0.83. However, the differences were small overall (Figure 6). Therefore, the survival outcomes of CUC patients in Bulgaria are still far from the survival outcomes in Western and Northern Europe [5].

## 5. Conclusions

Bulgaria is still the only country among the current member states of the EU that does not have a screening program for CUC. Implementing screening programs for CUC can dramatically change epidemiology, and it is the most effective approach in the short and long term, as has been shown in countries with well-established population-based screening, where CUC is nowadays a rare disease. The situation in Bulgaria is very different and this is also affecting patients’ quality of life and survival. Without a screening program and without increasing the vaccination coverage, the numbers of new cases of CUC in Bulgaria will remain high. Only an increase in vaccination coverage, without establishing screening, could have a positive impact on new incidence rates in the future.

## Figures and Tables

**Figure 1 healthcare-11-00318-f001:**
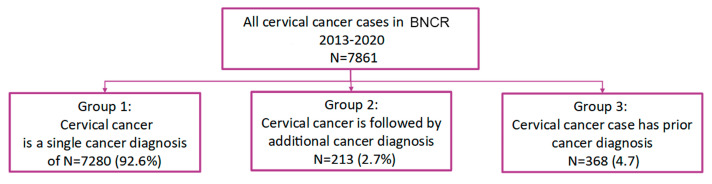
Patients according to number and time of diagnosis of cancer.

**Figure 2 healthcare-11-00318-f002:**
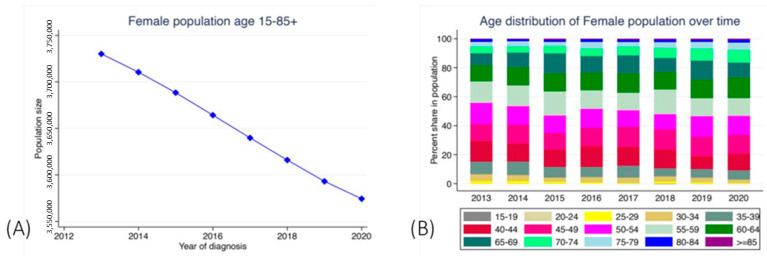
Distribution of women population with CUC according to (**A**) year and (**B**) age group.

**Figure 3 healthcare-11-00318-f003:**
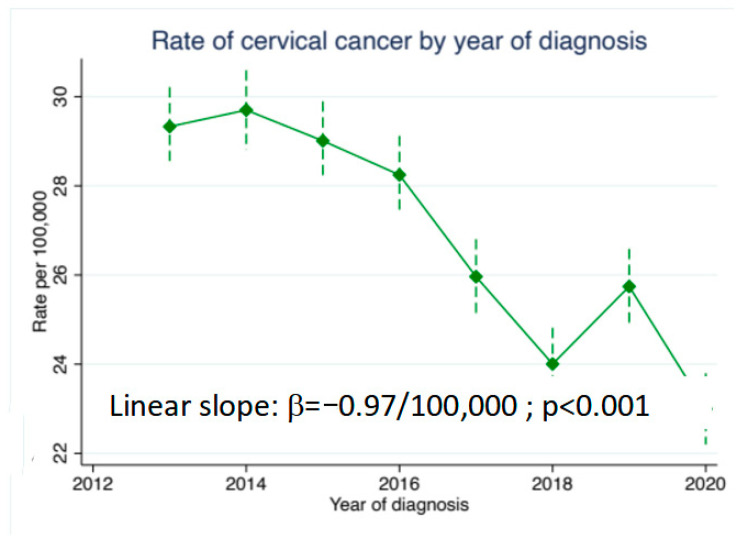
Rate of cervical cancer according to year of diagnosis. Average decline of −0.97/100,000 per year; *p* < 0.001.

**Figure 4 healthcare-11-00318-f004:**
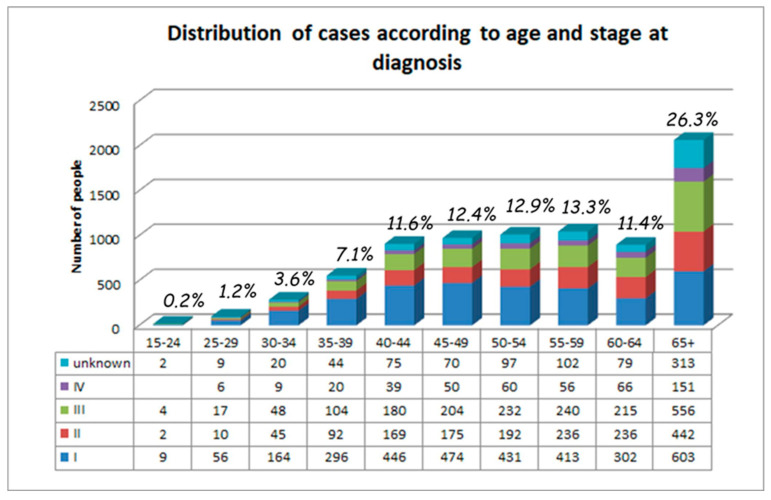
Distribution of CUC cases according to age and stage of diagnosis.

**Figure 5 healthcare-11-00318-f005:**
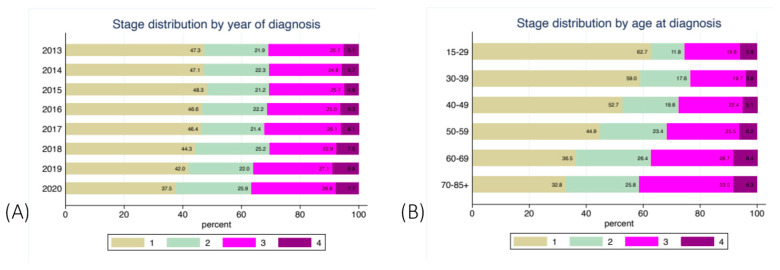
Distribution of stage of CUC (excluding those with unknown stage) by year (**A**) and by age (**B**).

**Figure 6 healthcare-11-00318-f006:**
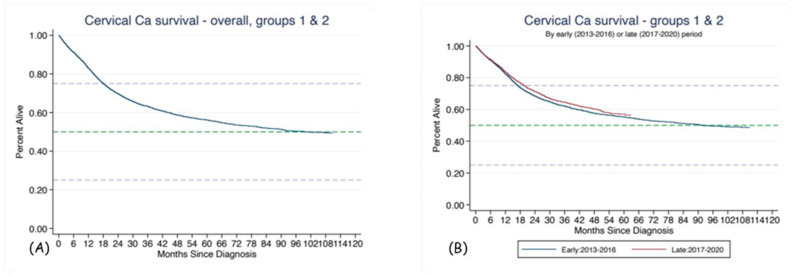
Overall survival of CUC patients from diagnosis: Groups 1 and 2 combined (**A**); according to year of diagnosis, early 2013–2016 vs. late 2017–2020 (**B**).

**Figure 7 healthcare-11-00318-f007:**
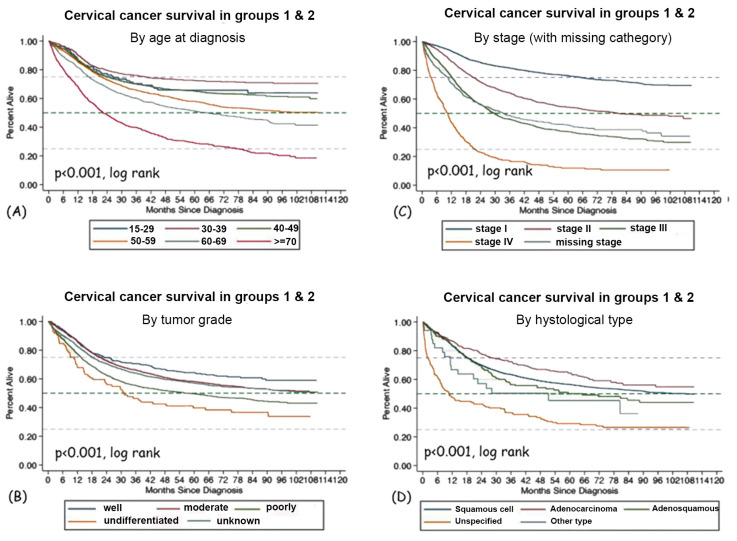
Survival according to different prognostic indicators: age at diagnosis (**A**); tumor grade (**B**); stage (**C**) and histological type (**D**).

**Figure 8 healthcare-11-00318-f008:**
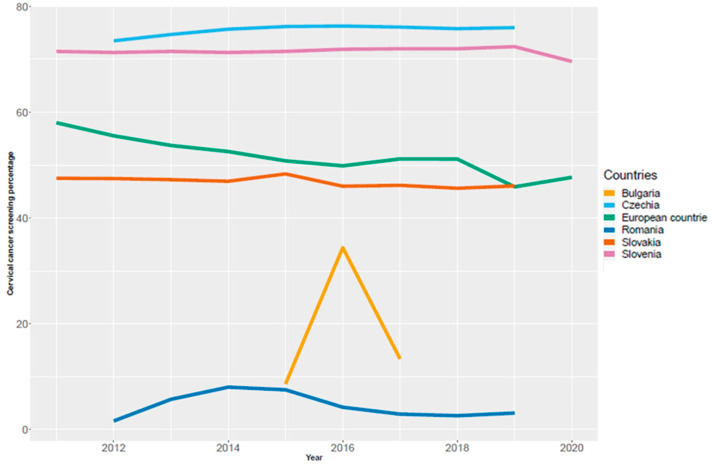
Coverage of Bulgarian population by screening program, 2013–2020 [20].

## Data Availability

Data available in a publicly accessible repository that does not issue DOIs. Publicly available datasets were analyzed in this study. This data can be found here: https://www.nsi.bg/en; https://infostat.nsi.bg/infostat/pages/external/login.jsf; https://ecis.jrc.ec.europa.eu/. Summary of the data used from the Bulgarian National Cancer Registry is publicly available here: https://www.sbaloncology.bg/index.php/bg/%D1%81%D1%82%D1%80%D1%83%D0%BA%D1%82%D1%83%D1%80%D0%B0/%D0%BD%D0%B0%D1%86%D0%B8%D0%BE%D0%BD%D0%B0%D0%BB%D0%B5%D0%BD-%D1%80%D0%B0%D0%BA%D0%BE%D0%B2-%D1%80%D0%B5%D0%B3%D0%B8%D1%81%D1%82%D1%8A%D1%80.html. The detailed data from the Bulgarian National Cancer Registry are available on request from the corresponding author since it is only accessible with permission from the BNCR.

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
