# Peer review of "Cancer of the Cervix in Bulgaria: Epidemiology of a Crisis"

_healthcare, 2023, doi:10.3390/healthcare11030318_

Round 1
Reviewer 1 Report
Comments
The study investigated the epidemiology, prognosis and survival of uterine cervix cancer (CUC) in Bulgaria from 2013 to 2020. The article is well throughout and signifies the 7 years trends of this cancer. With some constructive criticism I would like to improve the quality suggesting a major revision of this research.
Major
The conclusion is a bit distracting and should be focused/related more on the findings of the study.
Abstract:
1. The author has mentioned that he investigated the trends of CUC in context of Covid-19 outbreak and others, but there is no part discussing Covid-19 or its impact, so better omit it.
Introduction
1. Adequate
Materials and Methods
1. P2L90 Figure 1 is missing.
2. Figure 2 caption should be “Distribution of women population with CUC according to (A) year and (B) age group.
Discussion
1. P6L172 what is ind?
2. P7L208 Chine?
Conclusion
1. The conclusion of the study should be in line with the findings in the result section. I would recommend the author to be more specific in accordance with the findings of the results.
References
1. Follow the referencing style as recommended in instruction for the authors carefully. The page number/range of few of the references are missing.
2. Check out the necessary correction required in page / issue Ref # 7,22,23
3. Ref 11 and 12 are the same.
Author Response
Dear Editors,
We are pleased to submit a revised version of the manuscript (healthcare-2122484) entitled “Cancer of the cervix in Bulgaria - epidemiology of a crisis”. We appreciate the reviewers’ judgment and have now modified the manuscript accordingly. We believe the manuscript has benefited as a result. The point-by-point reply to reviewers’ comments are shown below.
Reviewer 1
Comments
The study investigated the epidemiology, prognosis and survival of uterine cervix cancer (CUC) in Bulgaria from 2013 to 2020. The article is well throughout and signifies the 7 years trends of this cancer. With some constructive criticism I would like to improve the quality suggesting a major revision of this research.
Authors’ reply: Thank you. We will do these suggested corrections.
The conclusion is a bit distracting and should be focused/related more on the findings of the study.
Authors’ reply: Thank you. We have amended accordingly.
Abstract: The author has mentioned that he investigated the trends of CUC in context of Covid-19 outbreak and others, but there is no part discussing Covid-19 or its impact, so better omit it.
Authors’ reply: Thank you very much for your suggestion. We added small discussion on COVID-19 in the text and removed it from the abstract. We think it is important since it has a real impact on mortality from any cause and it seems to have even great impact in Bulgaria. Still, this is not the main idea of our paper, so we prefered just to be mentioned, without focusing on it.
Materials and Methods
-
P2L90 Figure 1 is missing.
Authors’ reply: Thank you. Figure 1 has now been inserted
2. Figure 2 caption should be “Distribution of women population with CUC according to (A) year and (B) age group.
Authors’ reply: Thank you. We have amended accordingly.
Discussion:
1. P6L172 what is ind?
Authors’ reply: Thank you. We have amended accordingly, so it now reads: ‘decreased in all’
2. P7L208 Chine?
Authors’ reply: Thank you. We have amended accordingly, so it now reads: ‘China’
Conclusion
1. The conclusion of the study should be in line with the findings in the result section. I would recommend the author to be more specific in accordance with the findings of the results.
Authors’ reply: Thank you. We have amended accordingly.
References
1. Follow the referencing style as recommended in instruction for the authors carefully. The page number/range of few of the references are missing.
2. Check out the necessary correction required in page / issue Ref # 7,22,23
Authors’ reply: Thank you. We have amended accordingly.
3. Ref 11 and 12 are the same.
Authors’ reply: Thank you. We have deleted the duplicate reference and renumbered the rest of the references.
Dear Reviewer 1, many thanks for the kind comments made on our manuscript, for the accurate revision and the useful suggestions.
Best Regards
Prof Jean Calleja-Agius

Reviewer 2 Report
The authors studied the CUC epidemiology in their country from 2013-2020. The manuscript is descriptive, however, the overall writing has some formatting issues, like typing mistakes, syntax error. I would suggest the authors revise the manuscript. Certain sentences require clarify eg. L142The percentage of patients in stage one, who are over 65 years at diagnosis, is two times lower, L149, those with prior cancer diagnoses have lower survival rates.
The author should mention the number of patients of group 1 and 2, patient number of each age group and 2013-2016, 2017-2020, especially for Fig. 6A, B and 7A, 7C.
Fig. 1 is missing.
The explanation and legend for Fig.3 are insufficient, I would recommend for high resolution of Fig.7, particularly the color curves and lines.
Author Response
Dear Editors,
We are pleased to submit a revised version of the manuscript (healthcare-2122484) entitled “Cancer of the cervix in Bulgaria - epidemiology of a crisis”. We appreciate the reviewers’ judgment and have now modified the manuscript accordingly. We believe the manuscript has benefited as a result. The point-by-point reply to reviewers’ comments are shown below.
Reviewer 2
The authors studied the CUC epidemiology in their country from 2013-2020. The manuscript is descriptive, however, the overall writing has some formatting issues, like typing mistakes, syntax error. I would suggest the authors revise the manuscript. Certain sentences require clarify eg. L142The percentage of patients in stage one, who are over 65 years at diagnosis, is two times lower, L149, those with prior cancer diagnoses have lower survival rates.
Authors’ reply: Thank you. We have amended accordingly. Line 142:..... is two times lower (33%) and line 149 we added (92.5 vs 101.3 months).
The author should mention the number of patients of group 1 and 2, patient number of each age group and 2013-2016, 2017-2020, especially for Fig. 6A, B and 7A, 7C.
Authors’ reply: Thank you. These are now shown in Figure 1
Fig. 1 is missing.
Authors’ reply: Thank you. We have now included Figure 1
The explanation and legend for Fig.3 are insufficient
Authors’ reply: Thank you. We have added explanation to figure 3.
I would recommend for high resolution of Fig.7, particularly the color curves and lines.
Authors’ reply: Thank you. We have updated all figures in better resolution
Dear Reviewer 2, many thanks for the kind comments made on our manuscript, for the accurate revision and the useful suggestions.
Best Regards
Prof Jean Calleja-Agius

Reviewer 3 Report
Overall, the whole structure of study is good. However, some corrections are recommended for providing clear information. Particularly, I listed the following comments in detail here.
Major concerns:
In abstract, the author needs to mention the ingredients of methods. Also, the finding of the assay could be added step by step based on material and method. Please change “aim” to “aimed” and also “COVID-19” to “Coronavirus disease 2019”
In introduction, the citations of the literature are not appropriate, and some sentences lack reference, for example “This association raises the opportunity for an effective prevention of the disease with HPV vaccination. In 2018, the World Health Organization (WHO) announced the “Call for action” for eradication of CUC as a public health problem.” And “According to the report of the HPV center of the International Agency for Research on Cancer (IARC) in 2021, while there is no available data for Bulgaria, the HPV burden in Eastern Europe indicated that 9.7% of women had been carriers of infection with HPV 16/18 at some point in their life. These are exactly the same subtypes of HPV that are related to 84.7% of invasive CUC. HPV vaccination program was introduced in Bulgaria in 2017. In 2019 6.0% of the girls in the targeted population had received the first dose.”. Finally the authors should be write the aim of study as a past tense throughout the text. Hence, change “is” to “was”.
In methods, the author needs to mention the ingredients of methods.
In the discussion, discuss your results before relating them to the results of other published work. Also, the author must step by step to come to the results and comparison with others. What is your conclusion? Hence, add a significant statement that must be structured as “what was offered by authors? Do the authors have more thoughts on this field?
Author Response
Authors' Response
Dear Editors,
We are pleased to submit a revised version of the manuscript (healthcare-2122484) entitled “Cancer of the cervix in Bulgaria - epidemiology of a crisis”. We appreciate the reviewers’ judgment and have now modified the manuscript accordingly. We believe the manuscript has benefited as a result. The point-by-point reply to reviewers’ comments are shown below.
Reviewer 3
Overall, the whole structure of study is good. However, some corrections are recommended for providing clear information. Particularly, I listed the following comments in detail here.
In abstract, the author needs to mention the ingredients of methods. Also, the finding of the assay could be added step by step based on material and method. Please change “aim” to “aimed” and also “COVID-19” to “Coronavirus disease 2019”
Authors’ reply: Thank you. We have corrected in the main text and deleted this in the abstact
In introduction, the citations of the literature are not appropriate, and some sentences lack reference, for example “This association raises the opportunity for an effective prevention of the disease with HPV vaccination. In 2018, the World Health Organization (WHO) announced the “Call for action” for eradication of CUC as a public health problem.” And “According to the report of the HPV center of the International Agency for Research on Cancer (IARC) in 2021, while there is no available data for Bulgaria, the HPV burden in Eastern Europe indicated that 9.7% of women had been carriers of infection with HPV 16/18 at some point in their life. These are exactly the same subtypes of HPV that are related to 84.7% of invasive CUC. HPV vaccination program was introduced in Bulgaria in 2017. In 2019 6.0% of the girls in the targeted population had received the first dose.”.
Authors’ reply: Thank you. We have inclided the references accordingly
Finally the authors should be write the aim of study as a past tense throughout the text. Hence, change “is” to “was”.
Authors’ reply: Thank you. We have amended accordingly.
In methods, the author needs to mention the ingredients of methods.
Authors’ reply: Thank you. This is outlined under the subsection: Data and statistical analysis
In the discussion, discuss your results before relating them to the results of other published work.
Authors’ reply: Thank you. We have amended accordingly.
Also, the author must step by step to come to the results and comparison with others. What is your conclusion? Hence, add a significant statement that must be structured as “what was offered by authors? Do the authors have more thoughts on this field?
Authors’ reply: Thank you. We have amended accordingly.
Dear Reviewer 3, many thanks for the kind comments made on our manuscript, for the accurate revision and the useful suggestions.
Best Regards
Prof Jean Calleja-Agius

Round 2
Reviewer 1 Report
The author has revised the manuscript and necessary observations have been addressed.
1. Spaces are missing in between few words; P1L26 (aretrospective), P9L331 (thenumber).
2. Spelling mistake; P7L23; attempte
3. Figure 8; Greece (yellow) is missing. The color coding shown for cervical cancer screening percentage is for 6 countries but the mentioned countries are 7. Omit the name or add it accordingly.
Author Response
Dear reviewer,
Thank you very much for your comments. We have corrected accordingly.
- Spaces are missing in between few words; P1L26 (aretrospective), P9L331 (thenumber) - corrected
- Spelling mistake; P7L23; attempte -corrected
- Figure 8; Greece (yellow) is missing. The color coding shown for cervical cancer screening percentage is for 6 countries but the mentioned countries are 7. Omit the name or add it accordingly. -corrected, we removed Greece from the figure.
Thank you one more time!
Reviewer 2 Report
There are mistakes still present in the text. The authors are advised to revise the manuscript. Ex. there are two group 1 in Fig.1, age 55-99 in Fig.7A, L311, L324, L223
Author Response
Dear reviewer,
Thank you very much for your comment!
There are mistakes still present in the text. The authors are advised to revise the manuscript. - we revised the manuscript one more time
Ex. there are two group 1 in Fig.1 -corrected
age 55-99 in Fig.7A - corrected
L311 - corrected
L324 - corrected
L223 - corrected
Thank you one more time!